# Barriers to Childhood Immunisation and Local Strategies in Four Districts in South Africa: A Qualitative Study

**DOI:** 10.3390/vaccines12091035

**Published:** 2024-09-10

**Authors:** James Michael Burnett, Nqobile Myende, Algernon Africa, Mercy Kamupira, Alyssa Sharkey, Janine Simon-Meyer, Lesley Bamford, Sufang Guo, Ashnie Padarath

**Affiliations:** 1Research and Implementation Science Unit, Health Systems Trust, Durban 4001, South Africa; nqobile.myende@hst.org.za (N.M.); algernon.africa@hst.org.za (A.A.); ashnie.padarath@hst.org.za (A.P.); 2Health and Nutrition Section, United Nations International Children’s Emergency Fund South Africa, Pretoria 0011, South Africa; mkamupira@unicef.org (M.K.); asharkey@princeton.edu (A.S.); jasimon@unicef.org (J.S.-M.); sguo@unicef.org (S.G.); 3Child, Youth and School Health Cluster, National Department of Health (South Africa), Pretoria 0187, South Africa; lesley.bamford@health.gov.za

**Keywords:** immunisation, demand, uptake, barriers, facilitators, zero-dose, children, South Africa

## Abstract

**Introduction:** In South Africa over the past 20 years, immunisation has saved countless lives as well as prevented illnesses and disabilities. Despite this, vaccine-preventable illnesses remain a danger. The demand for and uptake of immunisation services are shaped by a variety of factors that can either act as barriers or facilitators to immunisation uptake. The aim of this project was to identify the supply and demand barriers and develop local strategies to improve childhood immunisation in four zero-dose districts in South Africa. **Materials and Methods:** This study used a mixed-method approach. In each of these four districts, 15 in-depth key informant interviews with health workers and local health managers and four focus group discussions (10 participants per focus group discussion) with community members and caregivers were held over a three-month period. Transcribed interviews were thematically analysed using qualitative analysis software (Nvivo^®^) into 10 factors as identified as important in influencing immunisation demand and uptake in previous studies. A further four were identified during the data analysis process. **Results:** Despite the varying role of factors affecting demand and uptake of immunisation services, three consistent findings stand out as major barriers across all districts. The first is *interaction with healthcare staff*. This clearly highlights the crucial role that the interactions between patients and staff play in shaping perceptions and behaviours related to immunisation services. The second is the overall *experience of care at healthcare facilities*. This emphasises the role that patient experience of services plays in perceptions and behaviours related to immunisation services. The third is *family dynamics*. This highlights the important role family dynamics play in shaping individuals’ decisions regarding immunisation uptake as well as the impact it has on the ability of people to access health services. **Discussion:** The role played by the different factors in the demand and uptake of immunisation services varied across the four districts examined in this study. Each of the districts presents a unique landscape where different factors have varying degrees of importance in affecting the utilisation of immunisation services. In some districts, certain factors are major barriers, clearly hindering the demand and uptake of immunisation services, while in others, these same factors might be a relatively minor barrier. This discrepancy highlights the unique nature of healthcare challenges across the districts and the need for tailored strategy recommendations to address them effectively.

## 1. Introduction

In 2024, the South African population stood at approximately 63 million, of which over a quarter (28%) were children younger than 15 years, making children the largest population group [1]. The majority of children (62%) are deemed as multidimensionally poor and suffering from various deprivations [2].

In South Africa over the past 20 years, immunization has prevented numerous illnesses and disabilities and has saved thousands of lives. However, despite these achievements, vaccine-preventable diseases remain a significant threat. The Expanded Programme on Immunisation in South Africa (EPI-SA) was established to safeguard children and pregnant women from preventable diseases through immunisation, aiming to prevent deaths and alleviate suffering caused by childhood illnesses.

The National Department of Health’s (NDoH) EPI-SA aims to achieve a 90% vaccination rate for all children [3]. From 2016–2018, 61% of children aged 12–23 months had received all recommended basic vaccinations, with 53% having received all age-appropriate vaccines. 

In 2019–2020, national immunization coverage for children under one year fell significantly short of the global benchmark of 90% [4]. There were notable disparities in coverage levels among provinces and districts. In response to a decline in immunization rates during the COVID-19 lockdown, the NDoH of Health launched a nationwide campaign from November 2020 to March 2021 aimed at catching up on childhood immunisations that cover children from birth up to 12 years of age [5].

Based on these experiences, the government acknowledged the need for a more targeted approach to implementing interventions to improve coverage, particularly at the local municipality and sub-district levels. South Africa comprises diverse settings where the reasons for zero-dose and under-immunized children vary significantly and are often intertwined with local contexts. Examining the underlying causes of these disparities across different health system levels, through the lens of the healthcare workers responsible for EPI service delivery, is essential. Equally important is understanding the demand-side obstacles from the community and client perspectives.

According to the literature, the demand for and uptake of immunisation services (for all age groups) is shaped by a variety of factors [6,7,8,9,10,11,12,13,14,15,16,17,18,19,20,21,22,23] that can either act as barriers or facilitators of immunisation uptake. A barrier is any obstacle or challenge that prevents individuals or communities from accessing or utilising immunisation services [6]. Barriers can be divided into major barriers (when 50% and more of references made by participants identify it as a barrier) and minor barriers (when less than 50% of references made by participants identify it as a barrier). A facilitator is any element that contributes to the success and utilisation of immunisation services. Certain aspects of individual factors present as both a facilitator and a barrier within the same district.

The conceptual framework used in this study categorises the factors influencing the demand and uptake of immunisation services into three distinct components, collectively responsible for the total process of accessing immunisation services. These components include the *demand side*, which is the multifaceted relationship between the individuals or communities seeking immunisation and the health system, the *service–delivery interface* through which vaccines are administered, and the *supply-side* infrastructure ensuring the availability and quality of vaccines [24,25]. Factors are aligned to the component where it is likely to take place. These components contain elements such as knowledge, beliefs, and attitudes towards vaccines on the demand side, the accessibility, availability, and acceptability of vaccination clinics on the service–delivery interface, and the efficiency, reliability, and adequacy of vaccine procurement, distribution, and storage on the supply side [24].

Evidence from the published literature relating to each of these is presented below. On the demand side, vaccine hesitancy has been widely cited as a key challenge. The WHO defines vaccine hesitancy as the “delay in acceptance or refusal of vaccines despite availability of vaccine services” [26]. The WHO has listed vaccine hesitancy as one of the top ten threats to global health with around 1.5 million vaccine-preventable deaths occurring each year [27]. Vaccine hesitancy is a complex challenge that can differ across time, location, and vaccines. 

Table 1 provides a summary of the various factors influencing immunisation uptake that have been identified in the published literature and aligns each factor with the three components of immunisation services.

The aim of this project was to identify the supply and demand barriers and develop local strategies to improve childhood immunisation in four zero-dose districts in South Africa. The specific objectives of this project were the following:To identify the demand-side barriers affecting the uptake of routine immunisation services from the perspectives of community members, health workers, and local health managers;To determine the service–delivery and supply-side factors affecting the uptake of services from the perspectives of community members, health workers, and local health managers;To identify context-specific community-focused strategies to improve routine immunisation services based on input from local stakeholders.

## 2. Materials and Methods

Insights were obtained from different stakeholders, such as caregivers, health workers, local health managers, to understand perspectives on the supply and demand barriers specific to these communities as well as their perspectives on what strategies might be important to reduce the barriers and improve immunisation coverage and equity.

This study used a mixed-method approach, which included a review of relevant policies and documents, key informant interviews (KIIs) with health workers and local health managers, and focus group discussions (FGDs) with community members and caregivers. Greater emphasis was placed on exploratory and descriptive information collected from KIIs and FGDs. Secondary quantitative data were used to better understand the demographic profile of the communities.

### 2.1. Study Sites

This study was undertaken in four zero-dose districts in South Africa, selected by the NDoH in collaboration with UNICEF. Table 2 provides an overview of the four study sites.

### 2.2. Study Participants

FGDs and KIIs were held with purposively selected stakeholders involved in and/or overseeing routine immunisation services. In each of the four districts, there were approximately 15 in-depth interviews with health workers and local health managers and four focus group discussions (10 participants per FGD) with community members and caregivers/parents from local healthcare facilities (see Table 3). Participants were identified using two recruitment methods. The first was the identification of a district focal person appointed by the respective district health manager who in turn assisted with the identification of health workers and local health managers overseeing routine immunisation services. The second recruitment method identified and invited eligible community members and caregivers/parents who are accessing immunisation services at a local healthcare facility. All participants were 18 years and older.

### 2.3. Data Collection

A desktop review was conducted to identify barriers affecting both the supply and demand for routine immunisation services as well as factors within the provision of these services that affect uptake of services. Zero-dose data were extracted from the District Health Information System (DHIS) as the secondary data source. Zero-dose children are children who have received the first dose of the diphtheria tetanus-pertussis-containing vaccine (DTPcv1) [31].

Face-to-face KIIs were conducted with participating health workers and local health managers (*n* = 60) using a semi-structured interview guide. Face-to-face FGDs were conducted with participating community members and caregivers/parents (*n* = 160) using a semi-structured discussion guide. The literature review [6,7,8,9,10,11,12,13,14,15,16,17,18,19,20,21,22,23] (and the identified factors and components of immunisation services) was used to help guide and inform the development of both the interview and discussion guides. KIIs and FGDs were conducted in English. The research team employed semi-structured guides to ensure comprehensive coverage of relevant topics of interest while also permitting spontaneity and flexibility for participants to introduce additional issues, thereby enhancing the data collection process [32,33].

Prior to participating in this study, potential participants received an Informed Consent Form (ICF) in plain language detailing this study’s objectives and emphasising voluntary participation. Informed consent was obtained from each participant. Key informant interviews (KIIs) and focus group discussions (FGDs) were conducted in English, with sessions recorded with consent and transcribed by research associates.

### 2.4. Data Analysis

A qualitative, thematic analysis of the data was undertaken using the qualitative data analysis software (Nvivo^®^ 14). Transcripts were carefully read by the research team. Data were analysed for themes relating to supply and demand barriers and community-driven strategies to address these barriers. Qualitative coding was used to identify and extract common ideas, patterns, or themes from the discussions. The findings were compiled using the 9 identified factors. However, to ensure the data analysis was robust and open to additional factors so as not to limit participants, an open theme was added during the analysis process, namely Other. This theme was used to further explore and seek additional factors which were then grouped into more factors in addition to the original 9 identified during the review process.

### 2.5. Ethical Clearance

The project was reviewed and approved by a recognised Independent Ethics Committee (IEC) and permission to conduct the project was obtained from the National Department of Health (NDoH) and respective Provincial Health Research Committees (PHRCs).

## 3. Results

In eThekwini Metropolitan Municipality, immunisation services face some challenges and barriers, but they are supported by two facilitators of immunisation, namely parental beliefs and gender equality. The four major barriers include interaction with healthcare workers, experience of care at healthcare facilities, accessibility of healthcare services, and family dynamics (see Table 4).

In the City of Johannesburg Metropolitan Municipality, district immunisation services are supported by one facilitator of immunisation, namely gender equality. The three major barriers reported include interaction with healthcare workers, experience of care at healthcare facilities, and family dynamics (see Table 5).

In Oliver Tambo District, immunisation services face six major barriers: religious beliefs, traditional health practices, interaction with healthcare workers, experience of care at healthcare facilities, accessibility of healthcare services, and family dynamics (see Table 6).

In Dr Kenneth Kaunda’s District, immunisation services face six major barriers affecting the demand and uptake of immunisation services: religious beliefs, traditional health practices, interaction with healthcare workers, experience of care at healthcare facilities, accessibility of healthcare services, and family dynamics (see Table 7).

Table 8 provides an overview of the facilitators, major barriers, and minor barriers per district.

## 4. Discussion

Immunisation is one of the most effective public health interventions, saving millions of lives annually by preventing infectious diseases. Despite the proven safety and efficacy of immunisation, several barriers affect the demand and uptake of immunisation services, and this is particularly true in so-called zero-dose communities.

The role played by the different factors in the demand and uptake of immunisation services varied across the four districts examined in this study. Each of the districts presents a unique landscape where different factors have varying degrees of importance in affecting the utilisation of immunisation services. In some districts, certain factors are major barriers, clearly hindering the demand and uptake of immunisation services, while in others, these same factors might be a relatively minor barrier. This discrepancy highlights the unique nature of healthcare challenges across the districts and the need for tailored strategy recommendations to address them effectively.

However, despite the varying role of these factors, three consistent findings stand out as major barriers across all districts. The first is *interaction with healthcare staff*. This clearly highlights the crucial role that the interaction between patients and staff plays in shaping perceptions and behaviours related to immunisation services. By improving the overall interaction with patients, healthcare workers can enhance trust, satisfaction, and ultimately, the utilisation of vital preventive services like immunisation. In essence, this finding highlights that beyond the medical aspects of healthcare, the patient’s interaction within the healthcare system significantly influences the demand and uptake of immunisation services. Addressing these factors is crucial for ensuring equitable access to immunisation and improving public health outcomes across diverse communities.

The second is that the *experience of care at healthcare facilities* presents as a major barrier across all districts. This emphasises the role that patient experience plays in shaping perceptions and behaviours related to immunisation services. On the other hand, negative experiences, such as long wait times, stock shortages, or inadequate communication, can deter community members from accessing immunisation services. By improving the overall experience of care, healthcare workers can improve satisfaction, and ultimately, the utilisation of vital preventive services like immunisation.

The third is that *family dynamics* present as a major barrier across all districts. This highlights the important role family dynamics play in shaping individuals’ perceptions, beliefs, and behaviours related to healthcare. Moreover, the influence of family members, particularly parents, on the decision-making process regarding vaccination cannot be overstated. The issue of teenage pregnancy coupled with the practice of parents leaving their children with their grandmothers in order to seek employment creates logistical challenges for accessing immunisation services, as the grandparents may prioritise childcare responsibilities over healthcare appointments. Grandparents left without Road to Health booklets also create a major barrier to their ability to access immunisation services. Therefore, understanding and addressing family dynamics are essential for effective public health interventions aimed at increasing immunisation coverage and preventing vaccine-preventable diseases.

It is interesting to note that the two metropolitan municipalities highlighted *gender equality* as a facilitator, whereas the two rural districts considered it a minor barrier. Another interesting finding was the reporting of *religious beliefs* as a major barrier in the two rural districts, whereas it is only considered a minor barrier in the metropolitan municipalities.

Strategy recommendations have been developed to address each of the factors that have been identified through the data analysis process. These recommendations have been crafted in a manner to help facilitate effective and locally acceptable approaches to address the barriers, not only in the study districts but also in other areas facing similar challenges with the uptake of childhood immunisation services. By aligning each recommendation with the corresponding component (namely *demand side*, *service–delivery interface*, and *supply side*) in the immunisation service process, this approach is intended to serve as a strategic tool for prioritising strategies, interventions, and allocating resources effectively, thereby working towards improving overall vaccine uptake and public health outcomes. Table 9 provides the recommendations and strategies in a concise format.

## 5. Conclusions

The role played by the various factors in the demand and uptake of immunisation services varied across the four districts, with each of the districts representing a unique landscape where different factors have varying degrees of importance in affecting the utilisation of immunisation services.

Despite the inherent differences observed between districts, the consistent emergence of three key factors as major barriers across all four districts creates a unique opportunity to capitalise on this commonality. It offers a strategic opportunity to develop a uniform intervention across district boundaries, streamlining efforts and maximising resource allocation. By recognising shared challenges, such as *interaction with staff*, *experience of care at healthcare facilities*, and *family dynamics*, stakeholders can devise a comprehensive intervention plan based on the strategy recommendations provided that address these common factors more effectively. Implementing a uniform intervention across all districts not only saves valuable time but also optimises the allocation of resources, avoiding duplication of efforts and ensuring equitable access to immunisation services for all communities. Leveraging this common ground fosters collaboration and knowledge-sharing among districts, facilitating the exchange of best practices and lessons learned. Through a unified approach, districts can collectively overcome these barriers and enhance immunisation coverage, ultimately advancing public health goals on a broader scale.

The recommendations have been crafted in a manner to help facilitate effective and locally acceptable approaches to address the barriers not only in the study districts but also in other districts or communities facing similar challenges with the uptake of childhood immunisation services. Further research could supplement and assist with validating the qualitative results of this study.

Childhood immunisation remains a cornerstone of global public health, offering substantial benefits in terms of disease prevention and community well-being. Addressing the barriers affecting the demand for and uptake of immunisation services requires sustained investment, innovation, and collaboration. The recommended strategies will assist districts in addressing these barriers. By prioritising childhood immunisation and implementing community-based strategies, we can ensure that every child has the opportunity to be protected from vaccine-preventable diseases.

## Figures and Tables

**Table 1 vaccines-12-01035-t001:** Summary of identified factors.

#	Factor	Refers to	Component of Immunisation Services
1	Trust in vaccine safety and efficacy	The level of trust held by individuals in the safety and efficacy of vaccines, and it plays a significant part in vaccine hesitancy and uptake. This can be both a general trust or a trust in a specific vaccination, both of which are related to a decrease in vaccine uptake [7,8].	Demand Side
2	Trust in the government and healthcare providers	The confidence that individuals have in the ability of governmental institutions and healthcare providers to make decisions that are in the best interest of public health and safety. This trust or lack thereof plays an important role in people’s willingness to utilise immunisation services [9,10,11,12].	Demand Side
3	Parental education, knowledge, and beliefs	Parents’ (paternal and maternal) education pertaining to healthcare services, their understanding of healthcare information (including the importance of immunisations, the role of vaccines in disease) as well as their attitudes and beliefs around immunisation. This plays an important role in the uptake of immunisation services. Parents who receive more healthcare information and related education tend to have better knowledge about vaccines and are more likely to utilise immunisation services [13,14,15,16,17].	Demand Side
4	Religious beliefs	A wide range of convictions, doctrines, and teachings from a specific religion that guide individuals’ understanding of the divine. Religious beliefs can impact vaccination decisions due to concerns about the ingredients of vaccines, the perceived interference with divine will, or religious exemptions from vaccination requirements [9,18,19,20].	Demand Side
5	Traditional health practices	The beliefs and rituals related to health that have been passed down through generations within a particular cultural group. Traditional health practices can include herbal medicine, spiritual healing, dietary guidelines, and various forms of traditional medicine. They are rooted in cultural beliefs, traditions, and knowledge systems and are often administered by traditional health practitioners (THPs). Research has shown that preference given to traditional medicine with a contempt for orthodox medicine reduces the acceptance and uptake of immunisation services [19].	Demand Side
6	Gender equality	The equal distribution of resources, opportunities, and decision-making power between men and women. In situations where there is reduced autonomy (for the mother), a partner’s refusal to consent to having children immunised has a negative impact on the uptake of immunisation services. In order to avoid domestic conflict, mothers are then also less likely to attend antenatal and immunisation services [14,21,24].	Demand Side
7	Accessibility of healthcare services	The ease with which individuals can access, reach, and utilise healthcare services. This includes geographical distance to the facility, availability of transportation, financial constraints, cultural and language barriers [14,19].	Service–Delivery Interface
8	Interaction with healthcare workers	The overall experience of individuals when dealing with healthcare staff. This can greatly impact their trust in the healthcare system and willingness to seek immunisation services. Perceived mistreatment and disrespect by staff is an important component in the decision-making process when a parent is deciding on whether or not to come for immunisation and antenatal services [9,13,14,22,23,24].	Service–Delivery Interface
9	Experience of care at healthcare facilities	Refers to the overall experience of individuals when visiting a facility. This is a multifaceted factor and includes (but is not limited to) quality of care received, stock shortages, and waiting times. Stock shortages can lead to delays in receiving services, increased frustration among parents and, ultimately, reluctance to go to the healthcare facility [8,24].	Supply Side

**Table 2 vaccines-12-01035-t002:** Study Sites.

District	eThekwini Metropolitan Municipality [28]	City of Johannesburg Metropolitan Municipality [29]	Oliver Tambo District Municipality [29]	Dr Kenneth Kaunda District Municipality [30]
Province	KwaZulu-Natal	Gauteng	Eastern Cape	North West
Population	3,900,000	5,500,000	1,514,306	809,441
Number of Households	1,125,765	1,680,000	314,079	240,544
% of Women-headed Households	42%	38%	57%	37%
% Child-headed Households	1%	0.3%	5%	0.40%
% Male Population	49%	50%	47%	49%
% Female Population	51%	50%	53%	51%
% Children (0–14 years)	25%	14%	46%	30%

**Table 3 vaccines-12-01035-t003:** Study Participants.

District	eThekwini Metropolitan Municipality	City of Johannesburg Metropolitan Municipality	Oliver Tambo District Municipality	Dr Kenneth Kaunda District Municipality
Number of KII with health workers and local health managers	15	15	15	15
Number of FGDs with community members and caregivers/parents	4 (40 participants)	4 (40 participants)	4 (40 participants)	4 (40 participants)

**Table 4 vaccines-12-01035-t004:** Facilitators and major barriers in eThekwini.

Factor		Result	Participant Quotation
Facilitators	Parental beliefs	There is a prevalent belief among participants that vaccinated children tend to meet developmental milestones more consistently compared to their unvaccinated counterparts.	*“…the immunised children are always on the right track and they walk faster as well”.*
	Gender equality	Fathers predominantly defer decision-making responsibilities to the mothers and grandmothers or are supportive of immunisation. This delegation of authority underscores the important role that women play in the immunisation process of children.	*“Yes, it was the mother of the child [who decides about immunisation]”.*
Major Barriers	Interaction with healthcare workers	Staff are described as rude with negative interactions between staff and patients.	*“You ask, but sometimes they shut you down rudely”.*
	Accessibility of healthcare services	Community members face significant challenges in accessing immunisation services due to a variety of factors such as long distances to facilities, shortage of reliable transportation, and financial constraints.	*“There are people from [inaudible] there, there is no transport. If you have one taxi that has left, it will not come back till the afternoon. Then you won’t be able to reach the clinic”.*
	Experience of care at healthcare facilities	The lack of direction within the facility on the flow and process, long waiting times, inconsistencies with staff work ethics, and stock shortages directly impact the experience of care by patients.	*“We get here very early and get service very late”.*
	Family dynamics	A prevalent occurrence is that of mothers leaving their children with grandparents or other elderly relatives. These grandparents often have difficulty accessing the local facility in order to access immunisation services.	*“Parents are working elsewhere. So they leave their child with, uh, their grandparents or aunts or uncles”.*

**Table 5 vaccines-12-01035-t005:** Facilitator and major barrier in City of Johannesburg.

Factor		Result	Participant Quotation
Facilitators	Gender equality	Gender equality appears very strong in the district and has significant influence over various aspects of family life, particularly in matters concerning the health of children.	*“…for me, because I know it was my teaching [what I was taught], but where I come from, my family values, we do immunisations. So I get it from my mother, it comes from family values and everyone, even me, I still have my card”.*
Major Barriers	Interaction with healthcare workers	District staff are referred to as rude and always shouting at patients. It has also been reported that staff refuse to speak English with foreign nationals.	“*Some are rude; they always shout at you*”.
	Experience of care at healthcare facilities	Long waiting times, inconsistencies with staff work ethics, stock and resource shortages, and load shedding all have a negative impact on the experience of care in the district.	“*It’s like you must take a day off, just to immunise your child* [referring to the long waiting times]”.)
	Family dynamics	The practice of parents leaving their children with their grandmothers in order to seek employment creates logistical challenges for accessing immunisation services. Furthermore, family members and a parent’s peers can play a role in deciding not to vaccinate their child.	*“What we’ve seen mostly is if people have homes in the rural area and because they know they’re working, all children are left to be cared for in the rural area by aunts, grandparents, et cetera, who will then not immunise the kids*”.

**Table 6 vaccines-12-01035-t006:** Major barriers in Oliver Tambo.

Factor		Result	Participant Quotation
Major Barriers	Religious beliefs	Some religions hold beliefs that are in conflict with certain elements of the vaccination programme, with certain churches actively opposing the immunisation programme.	*“Yeah, some of the churches discourage it. You know, if you go to that church, you don’t send a child for immunisations. You don’t. God is going help the child”.*
	Traditional health practices	It has a strong influence on healthcare perceptions and behaviour related to immunisations.	*“Some then believe on the traditional healers. So if the traditional healer, traditional healer does not command the person or the clients to go and immunise children, they don’t go. So I think it’s traditional”.*
	Interaction with healthcare workers	District staff are referred to as rude and always shouting at patients.	*“Sometimes you’ll find that they’ve written a date here, a date that it will be on the weekend. Then when you come here, you are telling them, you’ve gave me a date that it’s on the weekend, but they still won’t understand that they’ve only shouted at you instead of correcting their mistake”.*
	Experience of care at healthcare facilities	Experience of care is flunked greatly by long waiting times, stock shortages, and inconsistencies with staff punctuality and starting times.	*“The long waiting and the long queues”.*
	Accessibility of healthcare services	Community members face significant challenges in accessing immunisation services, with several factors contributing, namely reaching immunisation services due to their remote locations, prioritising immunisation over work responsibilities, and the actual cost of reaching the facility.	*“Almost three, three hour walk for you [to reach the clinic]”.*
	Family dynamics	The issue of teenage pregnancy often leads to seemingly disinterested parents who do not adhere to immunisation schedules. Furthermore, the practice of parents leaving their children with their grandmothers in order to seek employment creates logistical challenges for accessing immunisation services.	*“Yes, they, they just leave, the young mothers, they leave, the children. Yeah”*

**Table 7 vaccines-12-01035-t007:** Major barriers in Dr Kenneth Kaunda.

Factor		Result	Participant Quotation
Major Barriers	Religious beliefs	Some religions hold beliefs that are in conflict with certain elements of the vaccination programme.	*“Oh, they just believe in God, okay. So, so they don’t, they don’t believe in the immunisation system in some churches”.*
	Traditional health practices	In the district, the prevalent role of THPs in the health-related decision-making of some community members is an important factor.	*“Others [traditional healers] will tell you that, no, you have a spiritual what, what. You should focus on taking the child to spiritual, uh, cleansing and what, what and leave the clinic, if you take the child to the clinic, she or he will die. So come to me, I will help the child”.*
	Interaction with healthcare workers	Patients report negative interactions with staff with rude behaviour. Shouting is often mentioned as a main component of the negative interaction.	*“I think, uh, all nurses, all nurses there, they [are] not passionate about their job because all they do is shouting and shouting and they don’t keep that confidentially”.*
	Experience of care at healthcare facilities	Patients reported three main items that directly impact the experience of care by patients, namely long waiting times, stock shortages, inconsistencies with staff punctuality and starting times.	*“Some people still don’t, they don’t like to access our facilities because of the long waiting time. Others are a little bit reluctant, they don’t want to be in long queues of the facilities”.*
	Accessibility of healthcare services	Community members face significant challenges in accessing immunisation services due to a variety of factors such as long distances to facilities, shortage of reliable transportation, and financial constraints.	*“For those, uh, places that are far from the, more than five kilometres from the clinic”.*
	Family dynamics	The practice of parents leaving their children with their grandmothers in order to seek employment creates logistical challenges for accessing immunisation services, as the new caregivers may prioritise childcare responsibilities over healthcare appointments.	*“Yes, so you will see, you’ll only see when the grandmother brings the baby to the clinic when the baby is sick, that moment this baby has not been immunised”.*

**Table 8 vaccines-12-01035-t008:** Facilitators, Major, and Minor Barriers per district.

Component	Factor	District			
		ETH	JHB	ORT	DrKK
Demand Side	Trust in vaccine safety and efficacy				Minor
	Trust in the government and healthcare providers			Minor	Minor
	Parental education, knowledge and beliefs	Facilitator/Minor	Minor	Minor	Minor
	Religious beliefs	Minor	Minor	Major	Major
	Traditional health practices			Major	Major
	Gender equality	Facilitator	Facilitator	Minor	Minor
	Family dynamics	Major	Major	Major	Major
	Substance use			Minor	Minor
	Foreign national attitudes and beliefs			Minor	Minor
	Safety and security	Minor		Minor	
Service–Delivery Interface	Accessibility of healthcare services	Major	Minor	Major	Major
	Interaction with healthcare workers	Major	Major	Major	Major
Supply Side	Experience of care at healthcare facility	Major	Major	Major	Major

ETH = eThekwini; JHB = City of Johannesburg; ORT = Oliver Tambo District; DrKK = Dr Kenneth Kaunda District.

**Table 9 vaccines-12-01035-t009:** Strategy recommendations per component of the immunisation process.

Component	Recommendations and Strategies	Additional Content
Demand Side	Establishment of ***Peer Support Networks*** for the following:	Parents/young mothers to share experiences and share information. Grandparents to access educational materials. Grandparents connect with other caregivers in similar roles.
Demand Side	Develop and implement community-based ***Outreach Programmes*** with a special focus on the following:	Safety and efficacy of vaccines. Identification of proper sources of information. Building/strengthening relationships with churches, mosques. Building/strengthening relationships with THPs. Reaching young mothers. Addressing substance abuse. Tailored strategies to reach undocumented foreign nationals. Reaching grandparents and other caregivers.
Demand Side	Conduct further ***Research*** aimed at identifying the factors (facilitators and barriers) affecting gender equality.	Explore the social, economic, political, cultural, and institutional dimensions affecting gender equality and how this impacts the immunization program. Develop a social programme/intervention designed to address the factors affecting gender equality and its impact on decisions related to health-seeking behaviour.
Demand Side	Update the facility-based ***Parental Education Programme*** with a special focus on the following:	Identification of official sources of information. Religious beliefs. Traditional health practices. Patient complaints process.
Demand Side	A ***Media Campaign*** (using traditional channels such as TV, Radio, and Newspaper as well as using channels such as Facebook, Instagram, TikTok, and X)) with a special focus on the following:	Safety and efficacy of vaccines. Identification of official sources of information. The role of religion in decision making. The role of traditional health practices in decision making. Patient rights and responsibilities. Addressing misinformation and commonly held beliefs or myths. Services to foreign nationals.
Demand Side	Conduct further ***Research*** regarding the family dynamics of patients with a focus on the following:	Identifying which components of family dynamics impact the uptake of immunisation service. Developing family interventions to address the identified components.
Service–Delivery Interface	Conduct ***Soft Skills Training*** (on interpersonal communication skills) for all clinic staff with a focus on the following:	Relating to others and building trust. Better and open communication. Addressing religious beliefs. Addressing traditional health practices. Improving interaction between staff and patients. Managing the complaints process.
Service–Delivery Interface	***Invest in and/or expand*** the following initiatives:	MomConnect Whatsapp Initiative (and ensure it utilises a Whatsapp platform and not an SMS platform). Mobile clinic programme. Employer education and outreach programme.
Service–Delivery Interface	Develop an ***Action Plan*** (by the District with the Province) to include the following:	Address waiting times and staff punctuality. Ensure sufficient complaint boxes and other mechanisms are in place. Routinely educate staff and patients on the complaints process.
Service–Delivery Interface	Conduct further ***Research*** regarding the interaction between healthcare workers and patients with a focus on the following:	Identifying the factors influencing the interaction between healthcare workers and patients. Developing soft skills training to improve the interaction. Identifying the barriers to implementing the soft skills training.
Supply Side	Develop an ***Action Plan*** (by the District with the Province) to include the following:	Address stock shortage and staff shortages.
Supply Side	Conduct a ***Procurement Processes Review*** to include the following:	Identify procurement and stock management challenges as well as good practices. Share good practices with other districts. Address identified challenges.

## Data Availability

The original contributions presented in this study are included in the article; further inquiries can be directed to the corresponding author. The raw data supporting the conclusions of this article will be made available by the authors on request.

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
