# Peer review of "Barriers to Childhood Immunisation and Local Strategies in Four Districts in South Africa: A Qualitative Study"

_vaccines, 2024, doi:10.3390/vaccines12091035_

Round 1

Reviewer 1 Report (Previous Reviewer 1)

Comments and Suggestions for Authors

The authors were responsive to reviewer comments and made important edits to the manuscript. The remaining issue that can be easily corrected is the use of "literature" without citations. The authors use "literature" to identify barriers and facilitators (lines 68-70) and then characterize them as major and minor. There is only 1 citation for all these terms but they are critical to the analytical structure. The authors should add several well-regarded publications as citations to support the major and minor barriers and facilitators. The authors highlighted "literature review" on line 150 but it's still not clear which lit review they review. There isn't a specific lit review described earlier in the text so this point should be clarified. 

Author Response

Reviewer 2 Report (Previous Reviewer 2)

Comments and Suggestions for Authors

I do not find the answers for my last recommendations: can you provide more strategies in detail to help figure out interaction with healthcare workers and family dynamics since they are two main shared major barriers, for example, the timeline, the estimated cost, and the potential challenges.

In addition, in the discussion, the authors mentioned that “in other areas facing similar challenges with uptake of childhood immunization services.” Can the authors provide one or two area names that show these similar challenges?

Author Response

This manuscript is a resubmission of an earlier submission. The following is a list of the peer review reports and author responses from that submission.

Round 1

Reviewer 1 Report

Comments and Suggestions for Authors

This is a paper about barriers and facilitators for childhood vaccination. The authors should include "childhood" in the title to make that clear and focus the introduction on childhood vaccination issues.

They should also specify the ages of interest. Does "children" encompass newborns to 18 years old? Line 75, does this statement pertain to barriers and facilitators for any age group, including adults, or children and which ages?

The authors should open the article with a paragraph that provides the context not only data for childhood vaccination and condense the epi data presentation.

Lines 78-80, what is the source for these major and minor barriers and what is included in major and what included in minor? These must be defined, especially as they show up later in Table 8 and are used to classify factors and are in the discussion section.

Are the major and minor labels relevant to the factors in Table 1? The authors need to explain how Tables 1 and 2 relate to each other and relate to the aims.

How are the barriers and facilitators related to the service components? It appears they have already identified factors that the literature says are important but the aims say the team is identifying factors. The aims could be about testing how relevant factors from the literature about childhood vaccination barriers and facilitators are in the South African context of the 4 districts.

Definitions beginning on line 30 should be integrated with Table 1.

Re: study participants, authors should add tables that show the number of participants by district for each of the 2 participant types (health worker/manager or by community member, caregiver or parent) and recruitment methods. Provide details about recruitment- conversations with co-workers, emails, flyers, social media posts? How was the interview developed and where did the questions come from? Any models or theories used? Was this the purpose of the lit review mentioned on line 194? Or did they use the barriers, facilitators and service components from Tables 1 and 2 to generate the questions? Provide details about the data analysis.

Did the team begin deductively using the questions? Did they read the transcripts and generate open themes and codes?

Beginning on line 299 for recommendations, the text implies the recommendations are for the 4 districts but Table 9 states the recommendations generally. The authors need to strike a balance in this recommendations discussion. Readers won't be interested in recommendations that only apply to the 4 districts. Anything this specific would be for a project report to the facilities in the districts. But they can't generalize from this limited qualitative data. So they need to interpret the findings and rewrite the discussion in a way that shows what the findings could mean for other areas facing similar challenges in vaccinating children and suggest what other research could supplement and validate these qualitative results. 

Author Response

Comment 1: "This is a paper about barriers and facilitators for childhood vaccination. The authors should include "childhood" in the title to make that clear and focus the introduction on childhood vaccination issues."

Response 1: "Childhood" added to the title. 

Comment 2: "They should also specify the ages of interest. Does "children" encompass newborns to 18 years old?"

Response 2: Zero dose children specifically refers to children who have not received even their first dose of a diphtheria tetanus-pertussis-containing vaccine. This normally takes place within 6-14 weeks old. This study explores the barriers to childhood immunisation across all age groups.

Comment 3: "Line 75, does this statement pertain to barriers and facilitators for any age group, including adults, or children and which ages?

Response 3: For all age groups added. - line 85

Comment 4: "The authors should open the article with a paragraph that provides the context not only data for childhood vaccination and condense the epi data presentation."

Response 4: Context added to the first paragraph. It is difficult to reduce the epi data as it builds to background of the study. - Line 42-44

Comment 5: "Lines 78-80, what is the source for these major and minor barriers and what is included in major and what included in minor? These must be defined, especially as they show up later in Table 8 and are used to classify factors and are in the discussion section."

Response 5. Major barriers are when 50% and more of references made by participants identify it as a barrier. Minor barriers are when less than 50% of references made by participants identify it as a barrier. - line 88-90

Comment 6: "Are the major and minor labels relevant to the factors in Table 1? The authors need to explain how Tables 1 and 2 relate to each other and relate to the aims."

Response 6: Table 1 (line 189) provides a summary of the various factors influencing immunisation uptake that have been identified in the published literature while Table 2 provides an overview of the identified factors when aligned with the three components of immunisation services. The three components of the immunisation service are used a the framework against which the results are presented in order to better structure the recommendations.

Comment 7: "How are the barriers and facilitators related to the service components? It appears they have already identified factors that the literature says are important but the aims say the team is identifying factors. The aims could be about testing how relevant factors from the literature about childhood vaccination barriers and facilitators are in the South African context of the 4 districts."

Response 7. We identified factors based on the existing literature. These were explored to see if they are relevant in the 4 districts. Then we also explored and identified other factors not identified by the literature through the thematic analysis of the results.

Comment 8:"Definitions beginning on line 30 should be integrated with Table 1."

Response 8: Included as requested.

Comment 9:"Re: study participants, authors should add tables that show the number of participants by district for each of the 2 participant types (health worker/manager or by community member, caregiver or parent) and recruitment methods. Provide details about recruitment- conversations with co-workers, emails, flyers, social media posts? How was the interview developed and where did the questions come from? Any models or theories used? Was this the purpose of the lit review mentioned on line 194? Or did they use the barriers, facilitators and service components from Tables 1 and 2 to generate the questions? Provide details about the data analysis."

Response 9: 15 KII with managers/health workers and 40 participants (community members) for FGDs took part per district. A table would use up space at it it the same across all 4 districts.- line 300-303

Participants were identified using two recruitment strategies. The first was the identification of a district focal person appointed by the respective district health manager who in turn assisted with the identification of health workers and local health managers overseeing routine immunisation services. The second recruitment strategy identified and invited eligible community members and caregivers / parents who are accessing immunisation services at a local healthcare facility. - line 304-310

A qualitative, thematic analysis of the data was undertaken using the qualitative data analysis software (Nvivo®). Data were analysed for themes relating to supply and demand barriers and community driven strategies to address these barriers. -line 332-335

Comment 10: "Did the team begin deductively using the questions? Did they read the transcripts and generate open themes and codes?"

Response 10: The qualitative coding was used to identify and extract common ideas, patterns or themes from the discussions. The original 9 factors (identified by the literature) was used a themes and additional themes (factors) were identified. - line 336-340

Comment 11:"Beginning on line 299 for recommendations, the text implies the recommendations are for the 4 districts but Table 9 states the recommendations generally. The authors need to strike a balance in this recommendations discussion. Readers won't be interested in recommendations that only apply to the 4 districts. Anything this specific would be for a project report to the facilities in the districts. But they can't generalize from this limited qualitative data. So they need to interpret the findings and rewrite the discussion in a way that shows what the findings could mean for other areas facing similar challenges in vaccinating children and suggest what other research could supplement and validate these qualitative results."

Response 11: While the recommendations were developed for the 4 districts, they have been written in such as way that the recommendations will be applicable to other areas facing similar challenges.

Reviewer 2 Report

Comments and Suggestions for Authors

This article paid highly attention to the zero-dose immunization communities in South Africa. The authors focused on four districts in South Africa including eThekwini Metropolitan, City of Johannesburg Metropolitan, Oliver Tambo District, and Dr Kenneth Kaunda District. The authors conducted a qualitative analysis of the data using the qualitative data analysis software. The results showed that four different regions have shared facilitators and major barriers such as interaction with healthcare workers and family dynamics. These four regions also have their own. I believe that the summary table (table 8) was very helpful and useful for readers to directly know the facts or the trends that are happening in those four districts, resulting in the zero-dose immunization. The authors proposed corresponding strategies to figure out those problems.  To figure out those problems, I believe that seeking the donations will also be very helpful.

In addition, can you provide more strategies in details to help figure out interaction with healthcare workers and family dynamics since they are two main shared major barriers, for example, the timeline, the estimated cost, and the potential challenges. 

Author Response

Comment 1: "

This article paid highly attention to the zero-dose immunization communities in South Africa. The authors focused on four districts in South Africa including eThekwini Metropolitan, City of Johannesburg Metropolitan, Oliver Tambo District, and Dr Kenneth Kaunda District. The authors conducted a qualitative analysis of the data using the qualitative data analysis software. The results showed that four different regions have shared facilitators and major barriers such as interaction with healthcare workers and family dynamics. These four regions also have their own. I believe that the summary table (table 8) was very helpful and useful for readers to directly know the facts or the trends that are happening in those four districts, resulting in the zero-dose immunization. The authors proposed corresponding strategies to figure out those problems.  To figure out those problems, I believe that seeking the donations will also be very helpful."

Response 1: Thank you for the positive feedback to the manuscript, it is greatly appreciated.

Comment 2: "In addition, can you provide more strategies in details to help figure out interaction with healthcare workers and family dynamics since they are two main shared major barriers, for example, the timeline, the estimated cost, and the potential challenges. "

Response2: These issues you mention would make for an excellent follow-up study looking at a costing model and challenges with implementing the recommendations of this study.

Round 2

Reviewer 1 Report

Comments and Suggestions for Authors

It appears the authors did not agree with most of the comments provided as very few changes were made. They provided responses to reviewer comments in their reply letter but few edits to manuscript. 

Author Response

Comment 1: It appears the authors did not agree with most of the comments provided as very few changes were made. They provided responses to reviewer comments in their reply letter but few edits to manuscript. 

Response 2: The authors appreciate the reviews and comments provided by the reviewers. The authors respectfully responded to comment made by the reviewers in Round 1.